# Mass Spectrometry Characterization of the SDS-PAGE Protein Profile of Legumins and Vicilins from Chickpea Seed

**DOI:** 10.3390/foods13060887

**Published:** 2024-03-14

**Authors:** Antonella Di Francesco, Michele Andrea De Santis, Aldo Lanzoni, Maria Gaetana Giovanna Pittalà, Rosaria Saletti, Zina Flagella, Vincenzo Cunsolo

**Affiliations:** 1Laboratory of Organic Mass Spectrometry, Department of Chemical Sciences, University of Catania, Viale A. Doria 6-I, 95125 Catania, Italy; antonella.difrancesco@unict.it (A.D.F.); aldo.lanzoni@phd.unict.it (A.L.); marinella.pitt@virgilio.it (M.G.G.P.); rsaletti@unict.it (R.S.); 2Department of Agriculture, Food, Natural Resources and Engineering (DAFNE), University of Foggia, Via Napoli 25, 71122 Foggia, Italy; michele.desantis@unifg.it (M.A.D.S.); zina.flagella@unifg.it (Z.F.)

**Keywords:** chickpea proteins, legumins, vicilins, SDS-PAGE, Orbitrap^®^ mass spectrometry

## Abstract

Chickpea (*Cicer arietinum* L.) seed proteins show a lot of functional properties leading this legume to be an interesting component for the development of protein-enriched foods. However, both the in-depth proteomic investigation and structural characterization of chickpea seed proteins are still lacking. In this paper a detailed characterization of chickpea seed protein fraction by means of SDS-PAGE, in-gel protein digestion, high-resolution mass spectrometry, and database searching is reported. Through this approach, twenty SDS gel bands were cut and analyzed. While the majority of the bands and the identified peptides were related to vicilin and legumin storage proteins, metabolic functional proteins were also detected. Legumins, as expected, were revealed at 45–65 kDa, as whole subunits with the α- and β-chains linked together by a disulphide bond, but also at lower mass ranges (α- and β-chains migrating alone). Similarly, but not expected, the vicilins were also spread along the mass region between 65 and 23 kDa, with some of them being identified in several bands. An MS structural characterization allowed to determine that, although chickpea vicilins were always described as proteins lacking cysteine residues, they contain this amino acid residue. Moreover, similar to legumins, these storage proteins are firstly synthesized as pre-propolypeptides (Mr 50–80 kDa) that may undergo proteolytic steps that not only cut the signal peptides but also produce different subunits with lower molecular masses. Overall, about 360 different proteins specific of the *Cicer arietinum* L. species were identified and characterized, a result that, up to the current date, represents the most detailed description of the seed proteome of this legume.

## 1. Introduction

During the last decade, the interest in grain legumes has increased because they play an important nutritional role in the diet of millions of people in developing countries, but also because they represent a valuable source of protein, calcium, iron, phosphorus, and other minerals in the diets of vegetarians [1,2,3]. In particular, the chickpea (*Cicer arietinum* L.) is considered to be unique because of its potential health benefits, which include reducing cardiovascular, diabetic, and cancer risks, and because of its high protein bioavailability, biological value, and well-balanced amino acid profile [4]. On the other hand, some chickpea proteins were found to evoke positive reactions in chickpea-sensitive individuals [5,6]. Particularly, according to the Osborne fractions classification [7], chickpea seed proteins may be classified into four different groups: albumins, globulins, glutelins, and prolamins [8,9].

Albumins represent about 8–14% of the total protein content and are the water-soluble components of the protein fraction. They are mainly enzymatic and metabolic proteins, with a high content of sulfur-rich amino acids. Albumins are compact globular units, ranging in molecular weight from 10 to 80 kDa, typically composed of two polypeptide chains with eight cysteine residues [10]. Albumins also comprise many antinutritional components, such as amylase and trypsin inhibitors. 

Globulins are soluble in salt solutions and constitute the most abundant group, being 55–60% of the total protein fraction. Globulins can be further distinguished by their sedimentation coefficient into three classes: legumins (11S; S = Svedberg Unit), vicilins (7S), and a third type (which is present in a minor amount), known as convicilins [11]. As reported in the literature, legumins (11S) are the most abundant globulin components. They are hexameric proteins (300–400 kDa), constituted by six 54–60 kDa subunit pairs which interact noncovalently [12]. Each legumin subunit is known to be a precursor polypeptide, putatively containing an acidic and basic part linked in a single polypeptide chain, which are subsequently proteolytically cleaved to produce the separate chains [13]: the heavy acidic α-chain with molecular masses (Mr) of about 32–40 kDa, and the light basic β-chain with an Mr of about 20 kDa [14,15]. The α-chain mainly consists of glutamic acid residues and usually shows a leucine with an N-terminal amino group, while the β-chain is rich in alanine, valine, and leucine and usually has a glycine with an N-terminal amino acid. These two chains are linked by a one or more disulphide bonds and, therefore, can be released under reducing conditions. In this respect, previous studies [16,17], based on SDS-PAGE analyses, have shown that the most abundant protein fractions of chickpea are focused at the 40 and 25 kDa regions that are likely related to the α- and β-polypeptide chains of the 11S precursor protein, respectively [18]. 

On the contrary, as reported in the literature, the vicilins of chickpeas do not contain cysteine residues and, therefore, cannot form inter-chain disulphide bonds. They should assemble in a trimeric protein form (Mr 150–200 kDa), constituted by monomers (Mr of about 50–80 kDa) linked together by non-covalent hydrophobic bonds [19,20,21]. The literature’s data report extensive proteolytic processing that may occur in some species, such as the pea, producing many different size fragment subunits [22,23]. Particularly, vicilin precursor polypeptides are assumed to contain up to two potential sites for proteolytic cleavage, generating three fragments, classified as α (from the N-terminal), β (from the central part of the precursor), and γ (containing the C-terminus). In fact, in pea extracts, SDS-PAGE bands probably related to the α + β subunit (36 kDa), β + γ subunit (32 kDa), α- (19 kDa), β- (14 kDa), and γ-alone subunits (13–16 kDa) were identified.

Convicilins represent the third minor group of storage proteins. They show an Mr of about 70 kDa and are often found to be trimers of about 210 kDa constituted by three convicilin molecules, or heteromeric trimers of convicilin and vicilin. 

Glutelins (cereal-like proteins) are soluble in diluted acid or alkali solutions and represent about 18–25% of the total protein fraction. Glutelins contain high levels of methionine and cysteine residues and are therefore important from a nutritional point of view. However, currently, this class of chickpea proteins have not been well-investigated.

Finally, prolamins are alcohol-soluble proteins, and represent the smallest fraction (3–7%) of the total proteins in chickpeas. Like their cereal counterparts, these proteins are characterized by a high proportion of proline and glutamine, but, until now, they have not been well-characterized.

Despite the high nutritional value of the chickpea, there is little information on it at its protein-level [24,25]. Information has been mainly obtained using a combination of chromatography, SDS-PAGE, and mass spectrometry, and suggests that the subunit molecular characteristics of globulins and albumins in chickpeas are comparable to those of other legumes. In the meantime, the publication of a draft version of the chickpea genome in 2015 [26,27] was expected to facilitate not only genetic improvement and breeding, but also in-depth proteomic studies. Very recently, an integrated multi-omics analysis, carried out on root tissues of four contrasting drought-responsive chickpea genotypes, was performed with the aim of unraveling the complex mechanisms regulating the drought response in chickpeas [28]. However, both the detailed proteomic investigation and the structural characterization of chickpea proteins from seeds, and particularly those of its globulins, are still lacking. Here, the characterization of legumins and vicilins from chickpea seeds by means of SDS-PAGE, in-gel protein digestion, high-resolution mass spectrometry, and database searching, is shown.

## 2. Materials and Methods

### 2.1. Chemicals

The chemicals here employed were of the highest purity commercially available and were therefore used without further purification. Tris-HCl, Ethylenediaminetetraacetic acid (EDTA), dithiothreitol (DTT), ammonium bicarbonate (AMBIC), iodoacetamide (IAA) and formic acid (FA) were provided from Aldrich (St. Louis, MI, USA). Modified porcine trypsin was obtained from Promega (Madison, WI, USA). Water and acetonitrile (ACN) (OPTIMA^®^ LC-MS grade) for LC-MS analyses were purchased from Fisher Scientific (Milan, Italy).

### 2.2. Extraction of the Protein Fraction from Chickpea Seeds

The chickpea samples of the Pascià genotype were provided by the Department of Agriculture, Food, Natural Resources and Engineering (DAFNE) of the University of Foggia. The extraction of chickpea grain storage proteins was carried out according to a protocol adapted from the literature [29,30,31], and from De Santis et al. [32]. Briefly, the protein fraction was obtained by suspending 100 mg of chickpea flour with 1 mL of an extraction buffer (50 mM of Tris–HCl, pH 7.8, 5 mM of EDTA, 0.1% 1,4-dithiothreitol) for 1 h at room temperature with constant stirring, and then centrifuging at 10,000× *g* for 30 min. The extracted total of the soluble proteins (metabolic and storage proteins, with the exclusion of prolamins) were dried in a speed-vacuum (SpeedVac SPD1030 System, Thermo Fisher Scientific, Waltham, MA, USA).

### 2.3. SDS-PAGE Analysis

Proteins were separated via SDS-PAGE according to the protocol reported in De Santis et al. [30]. Separations with 12% polyacrylamide gels (SE 600, Hoefer Inc., Hollinston, MA, USA) were followed by gel staining with Coomassie BB G250 and were then digitally acquired (Epson Perfection V750 pro, Seiko Epson Corporation, Suwa, Japan). Four replications of the SDS-PAGE were carried out to assess the reproducibility of the protein profiles. Then, each of the selected bands was manually excised in duplicate from two lanes of the SDS-PAGE. Finally, the duplicate gel pieces were pooled, and transferred to a sterilized microcentrifuge tube (volume: 1.5 mL).

### 2.4. In-Gel Protein Digestion

The gel pieces were washed three times with sequential steps of water and acetonitrile. Then, they were subjected to the procedure of reduction and alkylation, followed by trypsin digestion, according to Shevchenko et al. [33]. This approach shows few modifications, as previously reported [34,35]. Briefly, to break the disulfide linkages in the proteins, the gel pieces were treated with 50 μL of 10 mM of DTT in 50 mM of AMBIC (pH 8.3) and incubated for 30 min at 56 °C. After the removal of the supernatant, the gel pieces were alkylated with 100 μL of 55 mM of IAA in 50 mM of AMBIC (pH 8.3) and incubated for 30 min at room temperature. After the removal of the supernatant, the gel pieces were washed with a 100 μL solution with a ratio 50:50 of water and ACN for 10 min, then the pieces were dehydrated with 100 μL of ACN. Finally, the in-gel enzyme digestion was carried out by adding 30 μL of 10 ng/μL of trypsin in ammonium bicarbonate (pH 8.3; 50 mM) for 30 min at 2–4 °C. After the removal of the supernatant, the gel pieces were incubated with 50 μL of 50 mM of AMBIC overnight at 37 °C. After in-gel digestion, the supernatant was transferred into a clean 1.5 mL tube. The gel pieces were treated with 50 μL of a 5% FA solution and subsequently with 50 μL of ACN to maximize peptide extraction. This procedure was repeated three times. At the end, the total extracts were pooled, joined with the first supernatant, lyophilized and re-dissolved in 20 µL of 5% FA for mass spectrometry analyses.

### 2.5. Mass Spectrometry Analysis

Mass spectrometry data were obtained using one microliter of the peptide mixture of each sample and acquired using the procedure described in Cucina et al. [36]. 

The analyses were carried out on a Thermo Fisher Scientific Orbitrap Fusion Tribrid^®^ (Q-OT-qIT) mass spectrometer (Thermo Fisher Scientific, Bremen, Germany) coupled online with the Thermo Scientific Dionex UltiMate 3000 RSLCnano system (Sunnyvale, CA, USA) liquid chromatography. The calibration of the MS system was performed using the Pierce^®^ LTQ Velos ESI Positive Ion Calibration Solution (Thermo Fisher Scientific), and MS data were acquired using Xcalibur v. 3.0.63 software (Thermo Fisher Scientific). Chromatographic separations were obtained using an Acclaim ^®^Nano Trap C18 Column (trapping column; 100 µm i.d. × 2 cm, 5 µm particle size, 100 Å) coupled online with a PepMap^®^ RSLC C18 EASY-Spray column (75 µm i.d. × 50 cm, 2 µm particle size, 100 Å). MS survey scans of the eluting peptide cation precursors were carried out at a high resolution (120 K resolution @ 200 *m*/*z*), whereas the tandem MS was performed via HCD fragmentation and acquired at a low resolution using the linear ion trap as the analyzer. To avoid carryover during nLC-MS/MS analyses, two blank runs were performed between two consecutive samples, and using the same sample gradient program.

### 2.6. Database Search Analysis

The MS data obtained were processed by PEAKS Xpro (release 20 October 2020; https://www.bioinfor.com//) de novo sequencing software (Bioinformatics Solutions Inc., Waterloo, ON, Canada) and searched against a dedicated protein database, including all the reviewed and unreviewed entries of *Cicer arietinum* L. (chickpeas) downloaded from the UniProt database (Swiss-Prot and TrEMBL sections, released February 2023, 31,239 entries). To identify the protein contaminants, MS data were also searched against the common Repository of Adventitious Proteins (c-RAP) contaminant database (www.thegpm.org). The following parameters were employed: (i) the tryspin was set as proteolytic enzyme; (ii) the number of allowed missed cleavage sites was set to 3; (iii) as variable amino acid modifications, the oxidation of methionine, the acetylation (N-terminal protein), and the transformation of N-terminal glutamine and N-terminal glutamic acid residue in the pyroglutamic acid form were considered; (iv) as a fixed modification, the carbamidomethylation of cysteines was set. The precursor mass tolerance threshold was 10 ppm and the max fragment mass error was set to 0.6 Da. Peptide Spectral Matches (PSMs) were validated using a Target Decoy PSM Validator node based on q-values at a False Discovery Rate (FDR) ≤ 0.1%. PEAKS score thresholds for PSMs were set to achieve FDR values below 0.1% for PSMs, Peptide sequences, and Proteins identified from each database search. A protein was considered identified if a minimum of two peptides (including at least a unique peptide) were matched, and the sequence coverage was ≥ 5%. For each SDS slice, a relative internal quantitative analysis was carried out to obtain the relative abundance of each protein identified. This result was obtained using the value of the “Sample Area” of each protein reported in the PEAKS panel of the identified proteins. Protein sample areas were calculated by the PEAKS Xpro (release 20 October 2020; https://www.bioinfor.com//) software using the total of all peptide features from the unique supporting peptides. The mass spectrometry proteomic data have been deposited to the ProteomeXchange Consortium via the PRIDE [37] partner repository with the dataset identifier PXD049382.

## 3. Results

The characterization of chickpea seed protein fraction by means of SDS-PAGE, in-gel protein digestion, high-resolution mass spectrometry, and database searching was carried out. Twenty SDS gel bands (see Figure 1) were cut and analyzed, as reported in the Materials and Methods Section. In each band, MS data revealed the co-migration of many protein components that, taking into account their relative abundance (derived by the MS data), were classified into three groups: the most abundant proteins (usually one, two, or, at most, three for each band), minor components, and very minor components (i.e., proteins in trace amounts). The most abundant proteins almost always showed a relative abundance higher than 30% and up to 95%, although in a few bands, the relative abundances were in the range of 16–28%. On the other hand, minor components usually displayed relative abundances in the range of 4–15%, whereas very minor components (or trace components) showed relative abundances always below 3%. 

Table 1 lists the most abundant proteins identified in each SDS band analyzed, including their relative abundance, sequence coverage, and the number of matched peptides. Overall, in nine bands, the most abundant constituents were storage proteins (i.e., legumins 11S, vicilins 7S, and albumins 2S), five bands showed metabolic proteins as main components, whereas the remaining six bands were mainly related to both metabolic and storage proteins.

### 3.1. Overall Results: The Importance of the Publication of the Draft Version of the Chickpea Genome

About 360 different proteins were identified by searching the UniProtKB database restricted to the taxonomy of *Cicer arietinum L.* species (about 31,000 reviewed + unreviewed entries). A result which, to date, represents the most detailed description of the chickpea proteome from a seed (a detailed list of all identified protein components and the relative peptides are shown in Appendix A, respectively). This result was possible thanks to the very recent publication of the draft version of the chickpea genome, and the consequent update of the protein sequence entries (i.e., about 24,000 out of 31,000 were published in 2017). In fact, the advances in genomic studies and the translations into the corresponding protein sequences represent fundamental steps to maximize accurate protein identifications traceable to the species of origin. It is important to highlight that the existence of protein sequence databases of the organism under investigation plays a fundamental role in mass spectrometry-based proteomic workflows, and has a strong impact on the results of the search. Proteomic analysis has, among its goals, the identification of as many proteins as possible with high confidence. Therefore, if most of the proteins of the sample are not present in the database (i.e., the database of organisms closes to the organism under study), the peptides from such proteins, and, therefore, the related MS spectra, can be matched incorrectly to other proteins in the database, leading to many false positive identifications, or they might simply remain unmatched [38,39].

### 3.2. The Most Abundant Proteins Identified in Each SDS-PAGE Band

The characterization of band N.1 (apparent Mr of 150 kDa) revealed the co-migration of 17 different proteins (Appendix A). However, the most abundant component (at a relative abundance of 78%) was an LIM domain-containing protein A-like isoform (theoretical Mr 102 kDa, Uniprot Accession Number A0A1S2Z626). This protein is involved in biological processes related to the stimuli of abscisic acid (ABA), a plant hormone important in the response to environmental stresses, including drought, soil salinity, cold tolerance, heat stress, etc. The most abundant protein identified in band N.2 (apparent Mr 100 kDa) was a ribonuclease (Acc. No. A0A1S2YD23, Mr of 108 kDa; at a relative abundance of 23%), although another twenty minor and trace protein components were also recognized. 

Two lipoxygenase isoforms (Acc. Nos. A0A1S2XBN2, at a relative abundance of 48%, and A0A3Q7YAD4, at a relative abundance of 28%) were also detected as main components in the adjacent band N.3 (apparent Mr 90 kDa), although thirty-six other very low-abundance proteins were identified. Band N.4, focused at apparent molecular mass of 70 kDa, was related to thirty-seven minor or trace protein components and one most abundant one, namely a seed biotin-containing protein SBP65 (Acc. No. A0A1S2XET4, Mr of 71 kDa; at a relative abundance 77%). This protein belongs to the late embryogenesis-abundant (LEA) proteins, a whole family of proteins upregulated during dehydration stress and highly abundant during the later stages of seed development, which gives the seeds the ability to tolerate drought. MS data obtained by the protein in-gel digestion of the N.5 (apparent Mr of 65 kDa) and N.6 (apparent Mr of 60 kDa) bands revealed the co-migration of many proteins: twenty-six in band N.5, and forty-three in band N.6, respectively. But only two were the most abundant components: the alpha-dioxygenase 1 (Acc. No. A0A1S2XH76, Mr of 74 kDa, band N.5, at 53%) and a vicilin-like protein (Acc. No. A0A1S2Y087, Mr of 69 kDa, identified in both bands N.5, at 24%, and N.6, at 95%). Another vicilin-like protein (Acc. No. A0A1S2YZ56, Mr of 82 kDa; at a relative abundance of 34%) was identified as one of the two main components in band N.7 (apparent Mr of 55 kDa), together with a protein disulfide-isomerase (Acc. No. A0A1S2YBR2, Mr of 60 kDa; at a relative abundance of 30%), and 56 other minor or trace protein components. It is interesting to note that, although chickpea vicilins are always described by previous studies as proteins lacking cysteine residues, three carbamidomethyl-cysteine residues were also characterized in this vicilin-like protein. To our knowledge, this represents the first direct characterization of cysteine residues in chickpea vicilins. Band N.8 (with an apparent Mr of 47 kDa) shows the presence of the vicilin-like protein (Acc. No. A0A1S2Y087) already detected in band N.6, as a main component, and thirty-seven other minor or trace proteins. Analogously, two main proteins were detected in band N. 9 (with an apparent Mr of 44 kDa), together with thirty-five other low-abundant components. The two most abundant components were a provicilin-like protein (Acc. No. A0A1S2XYZ0, Mr of 65 kDa; at a relative abundance of 56%) and a sucrose-binding protein-like (Acc. No. A0A1S2XVJ8, Mr of 54 kDa; at a relative abundance of 32%). Again, some peptides of the provicilin-like protein carrying carbamidomethyl-cysteine residues were identified and characterized. Two globulins and the alcohol dehydrogenase 1 (Acc. No. A0A1S2YBZ6, Mr of 41 kDa; at a relative abundance of 18%) were the most abundant components in band N.10 (with an apparent Mr of 41 kDa), together with forty-one minor proteins. Particularly, the two globulins were a legumin J (Acc. No. A0A1S2XVG1, Mr of 60 kDa, at a relative abundance of 33%) and a basic 7S globulin-like protein (Acc. No. A0A1S2XV08, Mr of 48 kDa, at a relative abundance of 29%). The same basic 7S globulin-like (at a relative abundance of 41%), and a different isoform of legumin J (Acc. No. A0A1S3E1N3, Mr of 63 kDa, at a relative abundance of 21%) were identified as the most abundant proteins in band N.11 (with an apparent Mr of 38 kDa), together with thirty-six other minor components. Bands N.12 and 13 (with an apparent Mr of 36 and 34 kDa) were mainly determined by legumin proteins, although a lot of minor components were also identified (see Appendix A). In detail, a legumin A-like (Acc. No. A0A1S2XSB9, Mr of 59 kDa, at a relative abundance of 81%) was the main component in band N.12, whereas the legumins with the accession numbers A0A3Q7XNW1 (Mr of 56 kDa, at a relative abundance of 60%) and Q9SMJ4 (Mr of 56 kDa, at a relative abundance of 27%), respectively, were identified as the most abundant components in band N.13. The NADPH-dependent aldehyde reductase 1 (Acc. No. A0A1S2YP40, Mr of 32 kDa, at a relative abundance of 38%) and a vicilin-like protein (Acc. No. A0A1S2XQR4, Mr of 52 kDa, at a relative abundance of 28%) were the most abundant components detected in band N. 14 (with an apparent Mr of 32 kDa). The oil body-associated protein (Acc. No. A0A1S2XCR9, Mr of 29 kDa, at a relative abundance of 29%) and the NADPH-dependent aldehyde reductase 1 (Acc. No. A0A1S2XVM2, Mr of 35 kDa, at a relative abundance of 23%) were the most abundant proteins in band N.15 (with an apparent Mr of 30 kDa). The oil body-associated protein belongs to a highly conserved plant group of proteins (i.e., oleosin) that are important for solubilizing seed fats and playing an important role in regulating the biosynthesis, metabolization, and mobilization of lipids during seed maturation and germination [40,41]. Two group members of the late embryogenesis abundant protein family (i.e., a dehydrin DHN3: Acc. No. A0A1S2Z0P8, Mr of 20 kDa; a LEA protein D-34-like isoform X1: Acc. No. A0A1S3DX03, 22 kDa) and a vicilin-like protein (Acc. No. A0A1S2YZ56, Mr of 82 kDa) were the most abundant proteins identified in band N.16 (with an apparent Mr of 24 kDa). Another vicilin-like protein isoform (Acc. No. A0A1S2XQ88, Mr of 51 kDa, at a relative abundance of 66%) was identified as the most abundant protein in band N.17 (with an apparent Mr of 23 kDa), whereas an albumin-2 (Acc. No. A0A3Q7K771, Mr of 26 kDa, at a relative abundance of 82%) was the main protein responsible in band N.18 (with an apparent Mr of 22 kDa). Finally, two legumins (i.e., legumin-A, Acc. No. A0A1S2XSB9, Mr of 59 kDa; legumin-J, Acc. No. A0A1S2XVG1, Mr of 60 kDa) and a P24 oleosin (Acc. No. A0A1S2XJM3, Mr of 21 kDa) were identified as the most abundant components in bands N.19 and 20 (with an apparent Mr of ~20 kDa).

## 4. Discussion

The SDS-PAGE analysis of the grain chickpea protein fraction, carried out under reduction conditions, shows a similar profile, with only a few differences from the results previously reported [42,43]. The molecular weight distribution of the components ranges from 20 to 130 kDa. Overall, while the majority of the bands and the peptides characterized were related to vicilin and legumin storage proteins, many other metabolic functional proteins (generally as minor components) were also detected. The visual inspection of the SDS-PAGE profile reveals that the most intense bands appear to be mainly focused in three mass regions, namely the mass ranges of 45–65 kDa, 30–37 kDa, and 20–25 kDa. The mass spectrometry characterization and database search allowed us to ascertain that these three mass regions were mainly related to six vicilins (Acc. No_s_. A0A1S2Y087, A0A1S2YZ56, A0A1S2XV08, A0A1S2XQR4, A0A1S2XQ88, and A0A1S2XYZ0), and five legumin components (Acc. No_s_. A0A1S2XSB9, A0A3Q7XNW1, A0A1S2XVG1, A0A1S3E1N3, and Q9SMJ4). However, all these storage proteins were also identified as trace components (i.e., with relative abundances below 3%) spread along almost all of the SDS-PAGE gel. In addition, two other vicilins, with the Acc. No_s_. Q304D4 and A0A1S2YKD9, were also identified, although always as components in traces (at relative abundances usually below 2%). The pro-vicilin Q304D4 was spread into almost all the SDS-PAGE slices investigated, and the vicilin A0A1S2YKD9 was instead identified in four bands at lower masses (see Appendix A).

### 4.1. Characterization of Vicilins

Firstly, it is interesting to note that the MS data allowed us to ascertain that, although chickpea vicilins were always described as proteins lacking cysteine residues [11], five out of the six entries of vicilins identified here as the most abundant components carry out this amino acid residue (as reported in the corresponding unreviewed sequence entries from the UniProtKB database). Particularly, seven cysteine residues from four vicilins were characterized (see Appendix A). In detail, three out of the eight cysteines for A0A1S2YZ56 (Appendix A), two out of twelve for A0A1S2XV08 (Appendix A), one out of two for A0A1S2XQ88 (Appendix A), and one out of five for A0A1S2XYZ0 (Appendix A) were characterized. On the contrary, the A0A1S2Y087 vicilin shows three cysteine residues, but none were characterized (Appendix A), whereas the A0A1S2XQR4 vicilin does not show cysteines (Appendix A). The observation of the presence of cysteines in the sequence of chickpea vicilins, and particularly of an odd number in the A0A1S2XYZ0 (five cysteine residues) and A0A1S2Y087 (three cysteines) vicilins, suggests the possibility of inter-chain disulphide bonding to form multimeric complexes. 

In addition, it is interesting to observe that, although these six vicilins show a theoretical molecular mass in the range between 82 and 48 kDa, in the SDS-PAGE they were mainly spread along the mass region between 65 (band N.5) and 23 kDa (band N. 17), and some of them were identified in several bands. The identification of a single vicilin in multiple bands could be related to a possible degradation during protein isolation and post-treatment, but also to indigenous isoforms, generated by post-translational proteolytic events, as already reported in other legumes [22,23,44]. In this respect, the predicted sequence of vicilin A0A1S2YZ56 has 699 amino acids (no information about the signal peptide is reported in the UniProtKB entry), including ten cysteine residues, and shows a theoretical molecular mass of about 81.9 kDa. However, this protein was detected as a main component in two bands focused at a lower molecular mass than those expected by its putative sequence. In particular, the MS data revealed the presence of peptides related to this protein in the SDS-PAGE region with an apparent mass of 55 kDa (band N.7) and in band N.16 (with an apparent mass of 24 kDa). In particular, in band N.7, peptides flanked between the amino acidic positions 289 and 660 were identified, whereas in band N.16, only peptides between the amino acidic positions 102 and 264 were characterized (Figure 2a and Appendix A).

Taking into account these results, it should be hypothesized that the vicilin A0A1S2YZ56 undergoes a post-translational proteolytic event producing two lower-mass vicilin polypeptides. The first one is derived from the C-terminal portion and has a molecular mass of about 50 kDa (band N.7); the second one, instead, is constituted by 250 amino acids of the N-terminal portion of the mature protein, and is focused in band N.16. Similarly, the basic 7S vicilin A0A1S2XV08 has 440 amino acids, including the peptide signal constituted by 19 residues. In its mature form, this protein should have an Mr of about 46.1 kDa, but it was identified in two bands, N.10 (41 kDa) and 11 (38 kDa), at lower masses. In both these bands, it was identified by a lot of peptides spread along the full-length sequence, but with the exclusion of the N-term fifty amino acids of its mature form (Figure 2b and Appendix A). 

Consequently, although it cannot be excluded that specifically band N.10 may be related to an entire protein, it can be hypothesized that these two bands were due to an isoform, truncated at the N-terminus. Also, vicilins A0A1S2Y087 (predicted to have a mature protein Mr of 66.4 kDa) and A0A1S2XYZ0 (predicted to have a mature protein Mr of 61.9 kDa) were identified in multiple bands. Particularly, A0A1S2Y087 was identified as the most abundant protein in bands N.5 (with an apparent mass of 65 kDa), 6 (60 kDa), and 8 (47 kDa), and as a minor constituent in bands N.7 (55 kDa) and 2 (100 kDa) (Figure 2c). The comparison of the sequence coverage patterns observed suggests that in bands N.5 and 6, the full-length sequence was probably focused, whereas bands N.7 and 8 were related to shorter isoforms, lacking about eighty amino acids of the N-terminal part (Figure 2c and Appendix A). Finally, the presence, as a minor component, of this vicilin in band N.2 may be related to its dimeric form. Likewise, in bands N.8 (47 kDa) and 9 (44 kDa) the vicilin A0A1S2XYZ0 (with a predicted Mr of 61.9 kDa) was identified. Taking into account that in both these bands many peptides were characterized as coming from the entire sequence, but that we never detected peptides belonging to the N-terminal part (about eighty amino acids, Figure 2d and Appendix A), it can be supposed that both bands N.8 and 9 are related to an isoform of the vicilin A0A1S2XYZ0, lacking this N-terminal set of amino acids. Finally, both the vicilins A0A1S2XQR4 (with a predicted Mr of 49.4 kDa, as a mature chain) and A0A1S2XQ88 (with a predicted Mr of 48.6 kDa, as a mature protein) were identified only in SDS regions with lower molecular masses than those expected by their putative sequence entries. Particularly, A0A1S2XQR4 was detected in band N.14 (32 kDa), whereas A0A1S2XQ88 was identified in bands N.16 (24 kDa, as a minor component) and 17 (23 kDa, as the most abundant protein).

Both these two vicilins were identified, not by peptides coming from delimited regions of the sequence, but by a lot of fragments spread along the entire sequence-length (Figure 2e,f, Appendix A). Consequently, at this stage, it could only be hypothesized that the identification of these two vicilins in bands at lower molecular masses than the predicted ones might be related to a possible degradation that occurred during their extraction and/or post-treatments. On the other hand, the co-migration of low-mass isoforms, formed through different post-translational proteolytic cleavages, and coming from both the N- and C-terminal portion of the precursor protein, cannot be rejected a priori.

Overall, both the presence of cysteines in the sequence of chickpea vicilins and the identification of vicilin fragments at lower molecular masses than those expected by putative sequence entries suggest that, similar to legumins 11S, these storage proteins are firstly synthesized as pre-propolypeptides with molecular masses ranging from 50 to 80 kDa. Then, the pre-propolypeptide might undergo proteolytic steps that cut not only the signal peptides but also occur at a midpoint between two cysteine residues, leading to different subunits which remain linked together by disulphide bonds, and, therefore, can be released and detected at lower molecular masses by an SDS-PAGE analysis carried out under reducing conditions. Indeed, as already reported, many vicilins, in some species, undergo extensive proteolytic processing that creates a high degree of heterogeneity in the subunit population [45]. 

### 4.2. Characterization of Legumins

Concerning the legumins, five different components of this kind of storage proteins were detected along the mass range comprised between the apparent mass of 41 kDa (band N.10) and 20 kDa (band N.20), two SDS mass regions typically assigned to the heavy α- (with molecular masses of about 32–40 kDa) and light β-chains (with molecular masses of about 20 kDa), respectively. However, two legumins were also revealed, as minor components, in bands focused at higher mass regions, probably due to migrating as whole subunits (i.e., with the α- and β-chains linked together by a disulphide bond). In this respect, the subunit legumin A-like A0A1S2XSB9 was detected as the most abundant component in bands N.12 (with an apparent mass of 36 kDa) and N.19 (20 kDa) (see Table 1), but also as a minor constituent of bands N.2 (100 kDa) and N.7 (55 kDa) (see Appendix A). The sequence entry of this legumin A-like protein, reported with the unreviewed status by the UniProtKB database, is constituted by 518 amino acids. The twenty-one first amino acids belong to the signal peptide. Consequently, the entire mature form of this protein is constituted by 497 amino acids (with an Mr of about 57.1 kDa) and, like the other legumins, consists of disulphide-linked acidic (α-chain) and basic (β-chain) polypeptides [13,14] (Appendix A). By analogy with the homologous legumins of the pea (*Pisum sativum* L.), the heavy acidic α-chain has presumably 312 amino acids (trait Phe^22^-Asn^333^), an Mr of about 36.9 kDa, and shows three cysteine residues [14]. In its native form, two out of the three cysteines of the α-chain form an intra-chain disulphide bond, whereas the third one is linked with a cysteine of the β-chain by an inter-chain disulphide bond. 

The light basic β-chain, which can be presumably generated by a post-translational proteolytic cleavage between the Asn^333^ and Gly^334^ of the whole legumin-A, is therefore constituted by 185 amino acids (C-term trait Gly^334^-Ala^518^ of the entire precursor A0A1S2XSB9), and has a molecular mass of about 20.2 kDa. Moreover, the light basic β-chain shows two cysteine residues; one of them is allowed to form the inter-chain link between α- and β-chains, whereas the second one is probably used to generate multimeric complexes. In band N.12, most of the MS/MS data matching this entry were attributable to the α-chain (Figure 3a and Appendix A), whereas few data were associated with the β-chain; on the contrary, the peptides detected in band N.19 were almost exclusively due to the β-chain, and only two MS/MS were related to the α-chain (Figure 3a and Appendix A). On the other hand, the MS data suggest that band N.7 (55 kDa) contains, as minor constituent, the entire subunit, whereas in band N.2 (100 kDa) the dimeric form of this subunit is probably present as minor component (Figure 3a and Appendix A). In this respect, it appears interesting that in band N.7, it was identified that the peptide DNGFEETIcm-CTAR (cm-C: carbamidomethylcysteine) contains the two last C-terminal amino acids (i.e., the aspartic acid and asparagine) of the α-polypeptide linked to the N-terminal peptide of the β-chain (see Appendix A).

Similarly, the legumin A0A3Q7XNW1 (with an Mr of 54 kDa, as an entire mature protein) was identified as a main component in band N.13 (with an apparent mass of 34 kDa), and as a minor component in bands N.8 (with a mass of 47 kDa) and 19 (20 kDa). The MS data obtained from band N.8 suggest the probable presence of the entire subunit, because a lot of peptides spread along the entire sequence-length were characterized (Figure 3b and Appendix A). Moreover, as observed in band N.7, the MS/MS data relative to band N.8 allowed us to identify the same peptide DNGFEETIcm-CTAR (which is shared between the two legumins A0A3Q7XNW1 and A0A1S2XSB9), which contains the C-terminal asparagine of the α-chain and the N-terminal glycine of the β-chain linked together (see Appendix A). Band N.13 was probably related to the α-chain, as almost all the MS/MS-matched peptides coming from the acidic polypeptide, and only four on their own, were due to the β-chain. Finally, the opposite sequence coverage pattern observed in band 19 (only three of the MS/MS were related to peptides of the α-chain) suggests the presence of the β-chain. In the same way, the identification of the legumin A0A1S3E1N3 (with an Mr of about 59.6 kDa, as a mature protein) in two different bands, namely bands N.11 (38 kDa) and, as a minor component, band N.19 (20 kDa), can be interpreted. The complementary sequence coverage patterns observed between these two bands (Figure 3c and Appendix A) suggest the migration of the α-chain in band N.11, and the presence of the β-chain in band N.19. The legumin Q9SMJ4 (with an Mr of 53.9 kDa, as a mature protein) was identified in band N.13 (34 kDa). Almost all the MS/MS data were from peptides of the α-chain, whereas only two were coming from the basic polypeptide (Figure 3d and Appendix A). Therefore, it can be supposed that the acidic polypeptide migrated in this band. Finally, the legumin J-like A0A1S2XVG1 (with an Mr of 57.7 kDa, in its mature form) was identified in band N.10 (41 kDa) as a main component, and in another three bands (i.e., N.17, 19, and 20, in the mass region of 23–20 kDa), as a minor component. The sequence coverage patterns observed in bands N.19 and 20 (Figure 3e and Appendix A) clearly suggest the presence of the β-polypeptide. On the contrary, the MS/MS data obtained from the other two bands (i.e., N.10, with an apparent mass of 41 kDa, and 17, with 23 kDa) belong to peptides spread along the entire sequence-length (Figure 3e and Appendix A). Moreover, it is interesting to note that in both these two bands, the peptide NGLEETIcm-CSAR, which contains the C-terminal asparagine of the α-chain and the N-terminal glycine of the β-chain linked together, was identified (see Appendix A). As a consequence, at this stage, it could only be hypothesized that the migration of this legumin J-like protein in these two bands might be related to a possible degradation which occurred during its extraction and/or post-treatments. However, the existence of lower-mass isoforms, formed through different post-translational proteolytic cleavages, which leave the acid and basic polypeptides joined together, cannot be excluded. It is important to consider that most of the identifications were carried out with the chickpea reference proteins present in the Uniprot, obtained from genomic sequences [46] of a limited number of genotypes (e.g., lines Castellana and ILC3279), whereas here we investigated the genotype Pascià. On the other hand, a polymorphism in chickpea seed traits is known [47], possibly also in seed storage proteins [48]. In this respect, genotypic variability could help to explain the differences in peptide sequences, also in relation to the post-translational cleavage position in legumins. Finally, post-translational modifications due to environmental conditions could also occur [49]. Particularly, environmental factors, including abiotic stress (such as drought) and nutrient availability, are reported to influence seed protein composition in the chickpea in terms of its accumulation rate and, then, in modifying the proportion of protein fractions, and so affecting the final protein composition [32]. Also, specific protein expressions (e.g., heat shock proteins) may occur, influencing the composition of the proteins [50]. 

## 5. Conclusions

During the past decade, the interest in the chickpea has increased because it represents an interesting source of protein for the development of non-gluten, protein-enriched ingredients. Therefore, the detailed characterization of its protein fraction, and particularly of its globulins (i.e., legumins and vicilins), which constitute its most abundant group, is needed. Here, we showed a detailed characterization of legumins and vicilins from chickpea seeds through the use of SDS-PAGE, in-gel protein digestion, high-resolution mass spectrometry, and database searching. 

The SDS-PAGE analysis, with a distribution of the components ranging from 20 to 130 kDa, shows a similar profile to results previously reported. Twenty SDS gel bands were cut and analyzed. In each band, MS data revealed the co-migration of many protein components, but each with very different relative abundances. Each band was generally due to one, two, or, at most, three of the most abundant proteins, although minor and trace proteins were also identified. Particularly, in nine bands, the most abundant constituents were storage proteins (i.e., legumins 11S, vicilins 7S, and albumins 2S), five bands showed metabolic proteins as the most abundant components, whereas the remaining six bands were mainly related to both metabolic and storage proteins. Overall, the MS characterization of twenty SDS gel bands allowed the identification of about 360 different protein components of *Cicer arietinum* L., which, to date, represents the most detailed description of the chickpea proteome from a seed.

The most intense bands appeared to be mainly focused in three mass regions, namely, the mass ranges of 45–65 kDa, 30–37 kDa, and 20–25 kDa, and were related to legumins and vicilins. As expected, legumins were spread along different SDS-PAGE mass regions. In fact, they were revealed as whole subunits (α- and β-chains linked together) but were also detected with the α- and β-chains migrating alone. The MS data suggested that vicilins, similarly to legumins, and as already reported in the pea, are firstly synthesized as pre-propolypeptides that may undergo proteolytic steps producing different subunits at lower molecular masses. Moreover, although chickpea vicilins were always described as proteins lacking cysteines, they instead contain an amino acid residue which, in some cysteine odd-number carrying vicilins, may be used to form inter-chain disulphide bonds and multimeric complexes. The further investigation, also in non-reducing conditions, of the amino acid sequences of these storage proteins may improve the knowledge about the probable proteolytic steps that they undergo and that determine the existence of the low-molecular mass polypeptides detected here. In addition, further studies in relation to genotypic and environmental variability will be carried out on the basis of the in-depth outcomes from the current investigation. Particularly, using a comparative proteomic analysis via a shotgun approach and label-free quantification (LFQ), the impact of different water regimes and a nitrogen supply on the protein fraction composition of two Italian chickpea genotypes will be investigated. 

## Figures and Tables

**Figure 1 foods-13-00887-f001:**
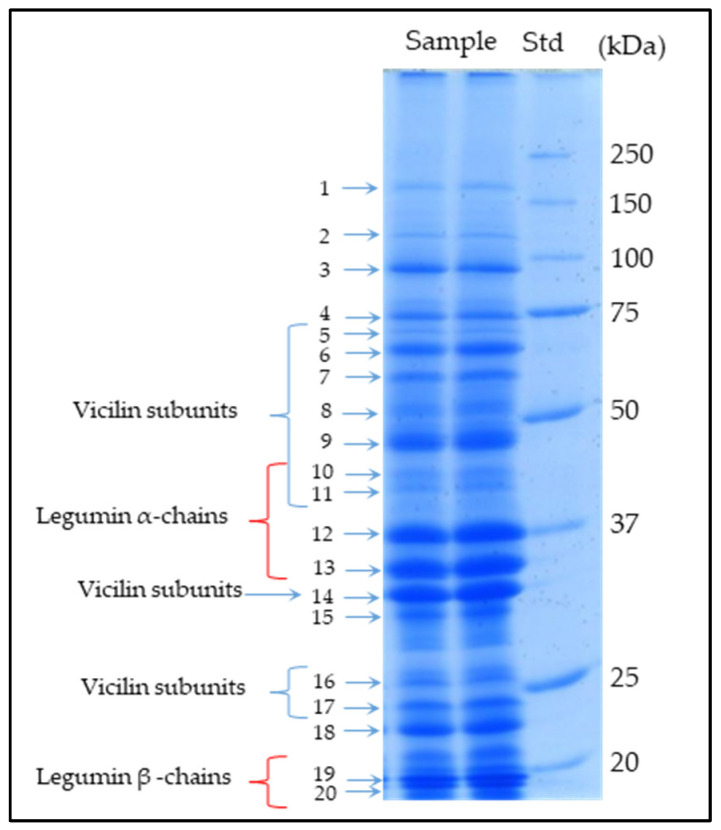
SDS-PAGE profile (two lanes as an example) of the chickpea seed protein fraction (genotype Pascià). Std: Standard protein markers. The twenty bands excised and analyzed via MS are shown by arrows. Bands containing 11S Legumins and 7S Vicilins as main components are labelled.

**Figure 2 foods-13-00887-f002:**
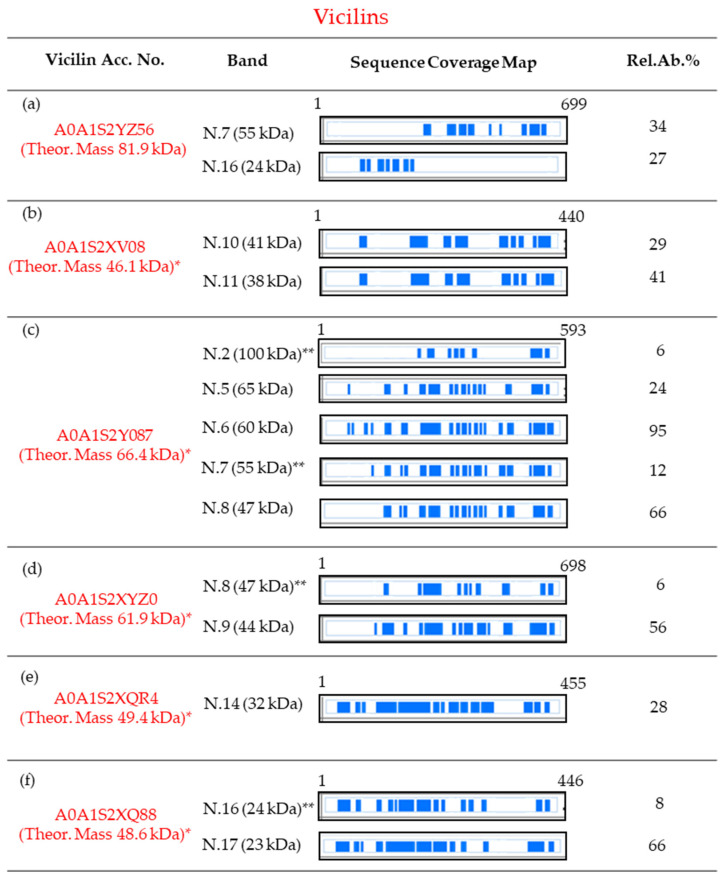
Protein sequence coverage maps of the vicilins identified as main, and in some case as minor, components in the SDS-PAGE slices of Figure 1. Information about each vicilin is reported in subfigures (**a**–**f**). Particularly, the coverage is visualized by blue blocks indicating the parts covered by high-confidence peptides (for the detailed sequence coverage pattern, see Appendix A). The column “Band” refers to the number of the band and the corresponding apparent electrophoretic mass; the column “Rel.Ab.%” shows the percent of its relative abundance in each band. Finally, the first and the last amino acid position are reported. Amino acid positions are referred to as the pre-propolypeptide sequences (i.e., containing the signal peptides). * Molecular mass calculated for the mature protein. ** Found as minor component.

**Figure 3 foods-13-00887-f003:**
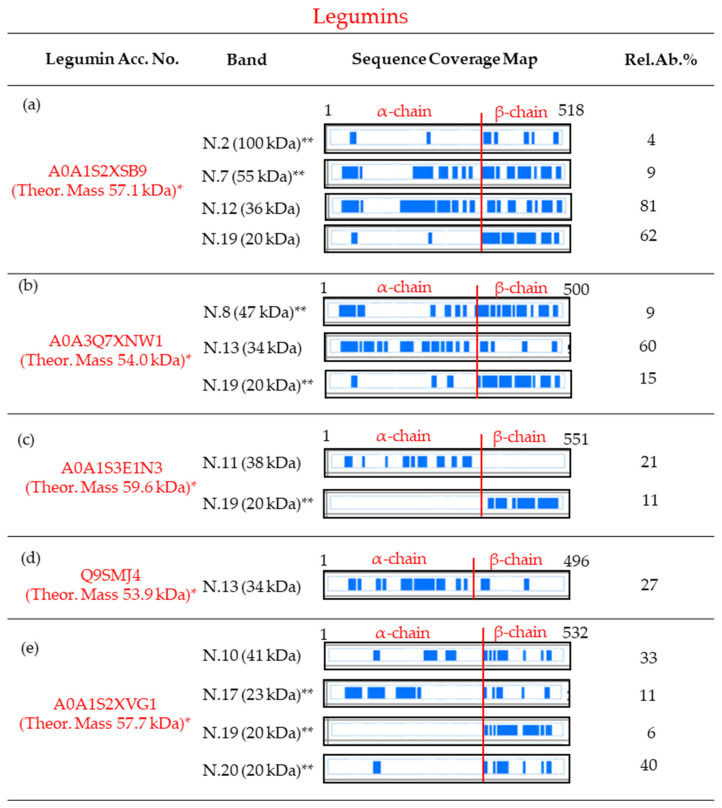
Protein sequence coverage maps of the legumins identified as main, or in some cases as minor, components in the SDS-PAGE slices of Figure 1. Information about each legumin is reported in subfigures (**a**–**e**). Particularly, the coverage is visualized by blue blocks indicating the parts covered by high-confidence peptides (for the detailed sequence coverage pattern, see Appendix A). The column “Band” refers to the number of the band and the corresponding apparent electrophoretic mass; the column “Rel.Ab.%” shows the percentage of its relative abundance in each band. Finally, the first and the last amino acid position are reported. Amino acid positions are referred to as the pre-propolypeptide sequences (i.e., containing the signal peptides). * Molecular mass calculated for the mature protein. ** Found as a minor component.

**Table 1 foods-13-00887-t001:** List of the most abundant proteins identified in each of the SDS bands analyzed (the number of the band and the apparent mass are shown). For each protein, the following are reported: the UniProt Accession number, average mass, relative abundance, PEAKS score, sequence coverage, number of matched peptides (P), number of unique peptides (UP), and the post-traslational modifications (PTMs) identified. All the protein entries identified are from the *Cicer arietinum* L. species.

Band No.	Mr SDS (kDa)	UniProt Acc. No.	Protein Description of *Cicer arietinum* L.	Av. Mass ^(a)^ (Da)	Rel. Abundance (%)	PEAKS Score(−10lgP) ^(b)^	Sequence Coverage (%) ^(c)^	P	UP	PTMs ^(d)^
1	150	A0A1S2Z626	LIM domain-containing protein A-like isoform X1	101,924	78	289.41	28	21	21	M^Ox^
2	100	A0A1S2YD23	Ribonuclease	108,276	23	272.17	29	23	23	M^Ox^; N-term Acetyl
3	90	A0A1S2XBN2	Lipoxygenase	97,493	48	369.20	48	43	31	Cm-C; M^Ox^; N-term Acetyl
		A0A3Q7YAD4	Lipoxygenase	96,937	28	363.07	46	36	25	Cm-C; M^Ox^; N-term Acetyl
4	70	A0A1S2XET4	Seed biotin-containing protein SBP65	71,295	77	398.02	64	62	62	Cm-C; M^Ox^; N-term Acetyl
5	65	A0A1S2XH76	Alpha-dioxygenase 1	73,557	53	304.08	42	24	23	Cm-C; M^Ox^; N-term Acetyl
		A0A1S2Y087	Vicilin-like	69,392	24	273.59	38	20	15	
6	60	A0A1S2Y087	Vicilin-like	69,392	95	358.89	49	41	36	
7	55	A0A1S2YZ56	Vicilin-like seed storage protein At2g18540	81,876	34	274.12	26	28	28	Cm-C; M^Ox^
		A0A1S2YBR2	Protein disulfide-isomerase	59,728	30	294.68	65	31	31	
8	47	A0A1S2Y087	Vicilin-like	69,392	66	317.44	40	28	23	
9	44	A0A1S2XYZ0	Provicilin-like	64,651	56	332.73	46	33	30	Cm-C; M^Ox^
		A0A1S2XVJ8	Sucrose-binding protein-like	53,803	32	363.89	58	39	39	Cm-C; M^Ox^
10	41	A0A1S2XVG1	Legumin J-like	60,371	33	250.18	30	16	16	Cm-C
		A0A1S2XV08	Basic 7S globulin-like	48,256	29	251.08	37	18	18	Cm-C; M^Ox^
		A0A1S2YBZ6	Alcohol dehydrogenase 1	41,069	18	213.96	40	15	15	Cm-C; M^Ox^; N-term Acetyl
11	38	A0A1S2XV08	Basic 7S globulin-like	48,256	41	287.79	36	15	15	Cm-C; M^Ox^
		A0A1S3E1N3	Legumin J	62,715	21	234.68	26	12	12	Cm-C; M^Ox^
12	36	A0A1S2XSB9	Legumin A-like	59,343	81	370.79	58	53	39	Cm-C; M^Ox^
13	34	A0A3Q7XNW1	Legumin-like	56,342	60	342.05	54	47	7	Cm-C; M^Ox^
		Q9SMJ4	Legumin	56,251	27	331.50	39	31	7	Cm-C; M^Ox^; N-term Acetyl
14	32	A0A1S2YP40	NADPH-dependent aldehyde reductase 1 chloroplastic-like	31,937	38	290.92	65	20	19	Cm-C; M^Ox^
		A0A1S2XQR4	Vicilin-like	51,839	28	361.96	66	65	11	
15	30	A0A1S2XCR9	Oil body-associated protein 2C	28,675	29	256.01	58	13	13	Cm-C; M^Ox^; N-term Acetyl
		A0A1S2XVM2	NADPH-dependent aldehyde reductase 1 chloroplastic-like	35,512	23	370.80	65	32	29	Cm-C; M^Ox^; N-term Acetyl
16	24	A0A1S2YZ56	Vicilin-like seed storage protein At2g18540	81,876	27	293.15	17	18	18	Cm-C; M^Ox^
		A0A1S2Z0P8	Dehydrin DHN3	20,553	27	258.00	45	14	7	M^Ox^; N-term Acetyl
		A0A1S3DX03	Late embryogenesis abundant protein D-34-like isoform X1	22,168	16	287.28	76	19	19	M^Ox^
17	23	A0A1S2XQ88	Vicilin-like	51,087	66	287.76	57	45	10	Cm-C
18	22	A0A3Q7K771	Albumin-2-like	26,148	82	308.09	85	22	16	Cm-C; M^Ox^; N-term Acetyl
19	20	A0A1S2XSB9	Legumin A-like	59,343	62	410.05	35	47	24	Cm-C; M^Ox^; N-term Acetyl
20	20	A0A1S2XVG1	Legumin J-like	60,371	40	240.63	17	10	10	Cm-C
		A0A1S2XJM3	P24 oleosin	20,683	34	212.25	35	9	9	

^(a)^ Molecular mass is calculated also including the mass signal, as reported by the UniProtKB database. ^(b)^ The PEAKS protein score (−10lgP) is calculated as the weighted sum of the −10lgP scores of the protein’s supporting peptides. ^(c)^ Sequence coverage is calculated considering the pre-polypeptide sequence (i.e., also including the mass signal), as reported by the UniProtKB database. ^(d)^ M^Ox^: Methionine Oxidation; cm-C: Carbamidomethylcysteine; N-term Acetyl: Acetylation of the N-terminal amino acid.

## Data Availability

The mass spectrometry proteomic data have been deposited to the ProteomeXchange Consortium (proteomexchange.org) via the PRIDE partner repository with the dataset identifier PXD049382. All ProteomeXchange datasets are publicly available. The datasets can be accessed via the individual repository websites. For PRIDE database, the datasets can be accessed via PRIDE Archive. In addition, datasets can be searched at ProteomeCentral (https://proteomecentral.proteomexchange.org/).

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
