# Peer review of "Mass Spectrometry Characterization of the SDS-PAGE Protein Profile of Legumins and Vicilins from Chickpea Seed"

_foods, 2024, doi:10.3390/foods13060887_

Round 1
Reviewer 1 Report
Comments and Suggestions for Authors
Proteomic investigation and structural characterization of chickpea seed proteins is important for functional studies of this bean. The paper uses an interesting method to conduct mass spectrometry on SDS gel bands. And vicilin and legumin storage proteins were confirmed in this bean. The results of the paper have implications for the development of chickpea protein.
I have a few questions and suggestions
Compared with traditional protein Mass Spectrometry Characterization, how does the "in-depth" in the title give more information?
Line 49-93 discusses various proteins respectively, which is a bit redundant. Please simplify it and merge these paragraphs.
Line 118 Is this extraction method able to extract all the functional proteins in chickpeas?
Is the symbol of the paper a problem of display, such as the division sign in 16-28% of line226?
Line 542-546 This part of the discussion about genome can be placed in the Results and Discussion section instead of the Conclusion section.
Also please simplify your conclusion. It is not recommended to cite references here. And please give some important data, such as the number of protein types, important protein content, etc.
Author Response
We thank the reviewer for his useful comments. Please see the attachment that provides a point-by-point response to the reviewer’s comments

Reviewer 2 Report
Comments and Suggestions for Authors
The manuscript goes around the analysis of the protein profile of Chickpea seed. The manuscript is well-written and has a good concept. The work is integrated by the introduction's elements, which enable us to assess the manuscript's placement within its context. The methodology is suitable for what it proposes to evaluate. The results are clear, and the discussion is well presented. The conclusion of the work is based on evidence from the work. In general, the manuscript represents an important point; however, minor remarks are suggested below:
Comments
1. Please add a separate list of abbreviations.
2. In section 2.4, please add a reference for each method used.
3. The last paragraph in the results section is too long. Please divide it into 2 or 3 paragraphs according to your results.
4. References should be updated, please.
Author Response
We thank the reviewer for his/her useful comments. Please see the attachment that provides a point-by-point response to the reviewer’s comments

Reviewer 3 Report
Comments and Suggestions for Authors
The current paper entitled “An in-depth Mass Spectrometry Characterization of the SDS-PAGE Protein Profile of Legumins and Vicilins from Chickpea Seedwork” by Francesko et al. (2024) is interesting, relevant, and sound. The writing and presentation of the facts are equally digestible. It is anticipated that this work would be a valuable example and contribution to the area of functional properties of food. That said, it is also important to highlight a few issues worthy of the authors’ attention before publication. Please refer to the points below.
- Lines 18, 104, 388, 401, 529, 540 - Avoid using we and our study.
- Please explain the significance of genomic studies concerning this objective.
- Please use the “most abundant” instead of “major” components.
- In the discussion part, please explain how environmental factors may influence the composition of proteins.
- Please highlight the future perspectives for this research, outlining the methodologies or aspects that will be explored to enhance the current understanding of chickpea proteins.
Comments on the Quality of English Language
Minor editing of English language required
Author Response

(The authors gave the same response as above.)
